# Urinary Dickkopf 3 Is Not an Independent Risk Factor in a Cohort of Kidney Transplant Recipients and Living Donors

**DOI:** 10.3390/ijms25105376

**Published:** 2024-05-15

**Authors:** Ulrich Jehn, Ugur Altuner, Lino Henkel, Amélie Friederike Menke, Markus Strauss, Hermann Pavenstädt, Stefan Reuter

**Affiliations:** 1Department of Medicine D, Division of General Internal Medicine, Nephrology and Rheumatology, University Hospital of Muenster, 48149 Muenster, Germany; ugur.altuner@ukmuenster.de (U.A.); lino.henkel@ukmuenster.de (L.H.); ameliefriederike.menke@ukmuenster.de (A.F.M.); stefan.reuter@ukmuenster.de (S.R.); 2Department of Cardiology I-Coronary and Peripheral Vascular Disease, Heart Failure Medicine, University Hospital Muenster, 48149 Muenster, Germany; markus.strauss@ukmuenster.de

**Keywords:** kidney transplantation, kidney donation, immunosuppression, dickkopf 3, uDKK3, allograft failure, biomarker

## Abstract

Urinary dickkopf 3 (uDKK3) is a marker released by kidney tubular epithelial cells that is associated with the progression of chronic kidney disease (CKD) and may cause interstitial fibrosis and tubular atrophy. Recent evidence suggests that uDKK3 can also predict the loss of kidney function in CKD patients and kidney transplant recipients, regardless of their current renal function. We conducted a prospective study on 181 kidney transplant (KTx) recipients who underwent allograft biopsy to determine the cause, analyzing the relationship between uDKK3 levels in urine, histological findings, and future allograft function progression. Additionally, we studied 82 living kidney donors before unilateral nephrectomy (Nx), 1–3 days after surgery, and 1 year post-surgery to observe the effects of rapid kidney function loss. In living donors, the uDKK3/creatinine ratio significantly increased 5.3-fold 1–3 days after Nx. However, it decreased significantly to a median level of 620 pg/mg after one year, despite the absence of underlying primary kidney pathology. The estimated glomerular filtration rate (eGFR) decreased by an average of 29.3% to approximately 66.5 (±13.5) mL/min/1.73 m^2^ after one year, with no further decline in the subsequent years. uDKK3 levels increased in line with eGFR loss after Nx, followed by a decrease as the eGFR partially recovered within the following year. However, uDKK3 did not correlate with the eGFR at the single time points in living donors. In KTx recipients, the uDKK3/creatinine ratio was significantly elevated with a median of 1550 pg/mg compared to healthy individuals or donors after Nx. The mean eGFR in the recipient group was 35.5 mL/min/1.73 m^2^. The uDKK3/creatinine ratio was statistically associated with the eGFR at biopsy but was not independently associated with the eGFR one year after biopsy or allograft loss. In conclusion, uDKK3 correlates with recent and future kidney function and kidney allograft survival in the renal transplant cohort. Nevertheless, our findings indicate that the uDKK3/creatinine ratio has no prognostic influence on future renal outcome in living donors and kidney recipients beyond the eGFR, independent of the presence of acute renal graft pathology, as correlations are GFR-dependent.

## 1. Introduction

The progression of chronic kidney disease (CKD) and its development from acute kidney injury (AKI) is highly variable and poorly understood. Consequently, predicting disease progression and individual risk stratification remains challenging but is crucial in the context of kidney transplantation (KTx). The most common approaches for risk stratification and prognosis estimation are still based on the analysis of glomerular filtration rate and albuminuria [1].

Recently, urinary dickkopf 3 (uDKK3) has emerged as a biomarker with potential for estimating the risk of CKD development from AKI or CKD progression courses [2,3]. DKK3, a secreted glycoprotein, is expressed by stressed renal tubular epithelial cells [4]. In particular, Federico et al. showed that DKK3 is co-localized with aquaporins 1 and 2, which mark proximal and distal tubules, respectively, while other compartments of the kidney lack DKK3 [5].

Under pathological conditions, DKK3 synthesis, which is suppressed in healthy adults, is reactivated, potentially leading to renal fibrosis and tubular atrophy [6]. DKK3 has been attributed to induce epithelial–mesenchymal transition (EMT) and to impair angiogenic competence after vascular injury [7]. Its involvement in Wnt/β-catenin signaling has been linked to interstitial fibrosis and tubular atrophy (IF/TA) induction, alongside its role in modulating local T-cell responses [4].

In CKD patients, uDKK3 levels are significantly elevated compared to healthy individuals [3]. Furthermore, in patients undergoing cardiac surgery, preoperative uDKK3 has been identified as an independent predictor of postoperative AKI and subsequent loss of kidney function [2]. Given the potential of uDKK3 to identify patients at high risk of CKD progression regardless of the underlying cause [8,9], its relevance in living kidney donation and KTx settings is of great interest. However, studies in this context remain limited. Schuster et al. have recently shown that uDKK3, three and twelve months after KTx, correlates with allograft function and can predict allograft function up to 36 months [10].

To verify and expand upon these findings in the kidney transplant setting, including the analysis of living donors (LDs), we investigated uDKK3 as a promising marker for predicting future (allograft) kidney function in our cohort of kidney transplant recipients and LDs.

On the one hand, we aimed to study uDKK3 kinetics in relation to kidney function before and after unilateral nephrectomy in LDs and to assess the prognostic value of uDKK3 in this context of rapid loss of kidney function, but without any underlying renal pathology in a healthy individual. On the other hand, we aimed to study uDKK3 kinetics in a cohort of KTx recipients undergoing renal allograft biopsy to determine the cause. As the kidney biopsies were performed on the same day as the urine sampling in the recipient group, we were able to investigate possible associations between uDKK3 levels and histological findings and to compare these results with the LDs, who can serve as controls for the KTx recipients.

## 2. Results

### 2.1. uDKK3 and Kidney Function in Living Kidney Donors

The mean age at donation was 54.3 (IQR 25.9, 69.8) years. The 82 living donors (LDs) showed a median uDKK3/creatinine (uDKK3/crea) pre-Nx ratio of 280 pg/mg (IQR 170, 460). As each LD was screened for renal disease and significant cardiovascular disease prior to donation, these values were used as baseline control values for healthy individuals.

Then, 1–3 days after Nx, the uDKK3/crea ratio increased significantly (*p* < 0.001) to a median of 1490 (IQR 580, 2870) pg/mg, a 5.3-fold increase. In parallel, the eGFR decreased significantly from 94.0 mL/min/1.73 m^2^ (SD ± 14.3) to 54.4 mL/min/1.73 m^2^ (SD ±10.5) (*p* < 0.001), a loss of 42.1%. In the following year, the eGFR resolved significantly (*p* < 0.001) to a mean of 66.5 mL/min/1.73 m^2^ (±13.5). This was accompanied by a significant decrease in the uDKK3/crea ratio to a median of 620 pg/mg (IQR 330, 1170) (*p* < 0.001). The uDKK3/crea ratio still remained significantly above the baseline from pre Nx (*p* < 0.001) (see Figure 1 for an overview).

Two and three years after donation, the eGFR remained constant without any statistically significant differences (two years: 68.5 (SD ± 15.6), three years: 64.3 (SD ± 14.0) mL/min/1.73 m^2^).

In the healthy individuals (LD day 0), the eGFR and uDKK3 did not correlate in the linear regression analysis (*p* = 0.207). The same was true for days 1–3 (*p* = 0.191) and 365 (*p* = 0.470) after kidney transplantation.

To test the impact of uDKK3 in those patients with a lower eGFR < 60 mL/min/1.73 m^2^, we performed linear regression analysis in this subset of donors. There was also no significant association in donors with an eGFR < 60 mL/min/1.73 m^2^, neither 1–3 days after Nx (*p* = 0.680, r = −0.0004, beta = −0.055) nor one year after Nx (*p* = 0.241, r = −0.002, beta = −0.234).

None of the LDs died or required renal replacement therapy during the follow-up. Donor baseline characteristics and outcome data are presented in Table 1.

### 2.2. uDKK3 and Kidney Function in Kidney Transplant Recipients

The KTx recipients included in our study had a mean age of 51.2 (SD ± 15.6) years. The median time since KTx was 39.8 months (IQR 8.7, 108.0). Of the 181 recipients, 37 had no acute histologic allograft pathology, i.e., bland histology or isolated chronic damage without acute inflammation. Their median uDKK3/crea ratio was 910 (IQR 510, 2360) pg/mg. In contrast, acute histologic allograft pathology was diagnosed in 144 recipient biopsies. It was defined as any treatment requiring diagnosis by the pathologists (rejection, infection, calcineurin-inhibitor (CNI)-induced nephrotoxicity, recurrent glomerulonephritis (GN)) and acute tubular injury due to ischemia reperfusion (IRI). These recipients showed a median uDKK3/crea ratio of 2050 (IQR 710, 6610) pg/mg, which was significantly higher compared to the 37 recipients without acute histologic pathology (*p* = 0.008).

Patients with acute histologic pathology had significantly worse eGFR values (33.7 (SD ± 16.3)) mL/min/1.73 m^2^ compared to patients without acute pathology (41.7 (SD ± 16.6)) mL/min/1.73 m^2^ (*p* = 0.010). The mean eGFR at biopsy for both groups combined was 35.3 (SD ± 16.6) mL/min/1.73 m^2^. Within one year, the eGFR improved slightly in both groups to 37.1 (SD ± 18.3) mL/min/1.73 m^2^ in the group with acute histologic pathology and to 42.9 (SD ± 16.9) mL/min/1.73 m^2^ in the group without acute diagnosis. At one year, the eGFR was not significantly different between both groups (*p* = 0.085).

For an overview, the relevant outcome parameters of the recipients are summarized in Table 2.

The uDKK3/crea ratio was statistically significantly correlated with the eGFR at biopsy in univariable linear regression analysis (*p* < 0.001, r= 0.000, beta −0.304). Regarding future renal function, we could not find an association when looking at the change in eGFR within one year (*p* = 0.289, r = 0.007, beta = 0.083). As a next step, we performed a multivariable linear regression analysis to adjust the uDKK3/crea ratio for the eGFR at biopsy. Although the uDKK3/crea ratio at biopsy was significantly correlated to eGFR values one year after biopsy in the univariable analysis (*p* = 0.002, r = 0.000, beta 0.232), there was no independent association between the uDKK3/crea ratio and eGFR one year after biopsy (*p* = 0.941, r = 0.000, beta 0.004) after adjustment for the eGFR at the time of biopsy by a multivariable linear regression analysis. Rather, the association was dependent on the eGFR at the time of biopsy (Table 3), and the eGFR at biopsy predicts the eGFR one year later but not uDKK3.

The histological diagnoses underlying the biopsies could be differentiated into eleven categories, as shown in Table 4.

uDKK3 levels for each histologic diagnosis are shown as a box plot in Appendix A. The corresponding eGFR is shown in Appendix A.

The diagnoses with the highest degree of tubulitis (≥T2) were TCMR and BKPyVAN. These diagnoses were associated with significantly higher uDKK3 levels compared to the other diagnoses, but also by a lower eGFR (Appendix A).

We performed a multivariable linear model to test whether different histologic diagnoses caused different uDKK3/crea levels or whether uDKK3 levels were dependent on corresponding eGFR levels (Appendix A). The analysis revealed that uDKK3 was dependent on renal function, not histologic diagnosis.

IF/TA was assessed unequivocally in 177 of the biopsied patients. A total of 122 (68.9%) showed mild IF/TA, 35 (19.8%) moderate, and 20 (11.3%) severe IF/TA. We evaluated the three different IF/TA stages for differences in uDKK3/crea ratio and further compared IF/TA, which reflects chronic tubular and interstitial damage, with acute tubular injury in patients with IRI (Appendix A). Patients with a high IF/TA degree had significantly higher uDKK3 levels compared to patients with low IF/TA (*p* = 0.003). Patients with acute tubular damage due to ischemia–reperfusion injury showed similar uDKK3 levels compared to patients with a high IF/TA degree (*p* = 0.665).

Similar to the future eGFR, the association of uDKK3/crea with higher IF/TA grade is also not independent of the eGFR. After adjustment for the eGFR at the time of biopsy using multinominal logistic regression analysis, the uDKK3/crea ratio remained a non-significant factor (Appendix A).

Of the 181 recipients, 11 (6.1%) died and 44 (24.3%) showed a definitive allograft failure during follow-up.

In order to calculate a cut-off that best discriminates between the superior and inferior eGFR, we performed a ROC analysis. Using the Youden index, a uDKK3/crea ratio of 1650 pg/mL was identified as the best-performing cut-off. Using this cut-off value, we performed a Kaplan–Meier analysis for further allograft survival in the follow-up of biopsy and uDKK3 measurement (Figure 2).

This analysis showed a significant association of a uDKK3 ratio > 1650 pg/mg with a significantly higher incidence of allograft loss. Since Kaplan–Meier analysis does not allow for adjustment, we applied multivariable Cox regression analysis to adjust the uDKK3/crea ratio for the eGFR.

This multivariable Cox regression analysis, adjusted for the eGFR at the time of biopsy, revealed the uDKK3/crea ratio as a non-significant independent factor predicting allograft loss (Appendix A).

Allograft survival, the degree of IF/TA and distinct histological pathologies (TCMR, BKPyVAN), and the eGFR at the same time and one year later were significantly correlated with the uDKK3/crea ratio in univariable analyses. Nevertheless, multivariable analysis excluded an independent association between the uDKK3/crea ratio and these parameters. According to our observations, the strong association of the uDKK3/crea ratio with the current eGFR in the recipient group rather suggests the predictive value of the uDKK3/crea ratio for future allograft function and outcome, which was negated by adjustment for the eGFR.

### 2.3. Comparison between Donor and Recipient uDKK3/crea Ratio in Relation to eGFR

Since LDs before Nx serve as healthy controls and experience an acute loss of function due to Nx, we compared uDKK3 levels in relation to the eGFR of the recipients with those of the LDs before and after Nx (see Figure 1).

In the healthy individuals prior to Nx (Figure 1, red dots), the uDKK3/crea ratio was significantly lower than in the recipients (Figure 1, blue dots) (*p* < 0.001). Then, 1–3 days after nephrectomy (LD), the uDKK3/crea ratio increased significantly (Figure 1, yellow dots) (*p* < 0.001) and was not significantly different from the uDKK3/crea ratio (*p* < 0.704) of the recipients at this time point. Conversely, the eGFR remained slightly higher 1–3 days after Nx (54.4 (SD ± 10.5) mL/min/1.73 m^2^) compared with the recipients (41.7 (SD ± 16.6) mL/min/1.73 m^2^, *p* < 0.001). Within one year after Nx, the eGFR significantly increased (*p* = 0.002); in parallel, the uDKK3/crea ratio significantly decreased (*p* < 0.001) (Figure 1, green dots). Taken together, the eGFR was significantly higher in our living-donor cohort at any time point compared to the recipients. Nevertheless, the uDKK3/crea ratio levels of LDs 1–3 days after Nx were comparable to those of recipients.

## 3. Discussion

This study investigated the role of uDKK3 in KTx recipients undergoing same-day kidney biopsy. The primary objective was to determine whether uDKK3 levels are associated with recent allograft function and can predict the future course of allograft function, as previously observed in CKD patients [11]. In addition, we investigated whether uDKK3 is associated with acute or chronic histological allograft pathologies, IF/TA, since it is known to be associated with fibrosis and tubular damage [5]. While uDKK3 has been barely studied in KTx recipients, Schuster et al. reported that higher uDKK3 values at 3 and 12 months after transplantation predicted lower estimated glomerular filtration rate (eGFR) values over 36 months [10]. However, it is important to note that Schuster et al. used serum creatinine as a marker for kidney function in their mixed linear models, rather than the eGFR.

Further, the role of uDKK3 was examined in LDs before and after the “artificial” loss of kidney function. Up until now, there has been no published data on the role of uDKK3 in living donors. We compared the uDKK-3/creatinine ratio in LDs without underlying kidney disease, but with an “artificial” rapid loss of approximately 50% of kidney mass due to nephrectomy, resulting in about 29–42% loss of renal function (eGFR values, as shown in Table 1) compared to the uDKK3/crea ratios of the kidney transplant (KTx) recipients. The outcomes and baseline characteristics of the living-donor cohort in this study are comparable to those observed in other studies [12,13].

In LDs, the uDKK3/crea ratio significantly increased by 5.3-fold within 1–3 days after Nx. The baseline level reflects the state of healthy adults, which is consistent with the results of other studies [8]. Over the course of one year, it decreased to a median level that was still significantly higher than the baseline (2.2-fold, *p* < 0.001), even in the absence of any underlying primary renal pathology, which seems contrary to observations by other authors who strongly attribute uDKK3 elevations to pathological conditions [5]. Schunk et al. observed a uDKK3/crea ratio above 471 pg/mg as a cut-off for an increased risk of acute kidney injury in patients undergoing cardiac surgery [2]. Since kidney donors in our study showed an elevated uDKK3/crea ratio, which was even higher one year after the donation of 620 pg/mL (330, 1170) without the further deterioration of renal function in the following two years (Table 1), uDKK3 does not seem necessarily to be a pathogenic factor.

It is important to note that in these patients, kidney function, as indicated by the eGFR, decreased by an average of 42.1% immediately after nephrectomy and then significantly improved by about ~19% during the following year. The course of the uDKK3/crea ratio followed an inverse pattern, suggesting a correlation between uDKK3/crea ratio levels and renal function in the living donors. In contrast, the uDKK3/creatinine ratio in kidney transplant (KTx) recipients was significantly higher compared to healthy individuals, which corresponds to the lower eGFR values observed in our cohort. The mean eGFR in the recipient group was significantly lower compared to the donors, paralleled by a ~5.5-fold increased uDKK3/crea ratio compared to healthy individuals.

In KTx recipients, the uDKK3/crea ratio positively correlated with the risk of future allograft failure in our cohort (Figure 2), which at first glance is consistent with the findings of Schuster et al. [10]. However, it is important to note that this association is not independent of the eGFR. Using multivariable Cox regression analysis with the eGFR as an adjustment factor, this study showed that the uDKK3/crea ratio alone was not a predictive factor for the risk of allograft failure. The KTx recipients in the cohort of Schuster et al. showed a median uDKK3/crea ratio of 771.5 pg/mg with an eGFR of 49.5 mL/min/1.73 m^2^ after one year [10]. Our patients had a slightly higher median uDKK3/crea ratio of 910 pg/mg with a slightly lower eGFR of 44 mL/min/1.73 m^2^. Therefore, the populations of both studies seem to be comparable.

Although the recipients from our cohort had a higher risk of allograft failure in univariable analysis with uDKK3/crea ratios above the distinct cut-off of 1650 pg/mg creatinine, eGFR estimation based on the widely used CKD-EPI formula remained a stronger predictor of allograft failure. A previous study by Zewinger et al. found that elevated uDKK3/crea levels above 4000 pg/mg predict eGFR decline in CKD patients over the next twelve months, regardless of the cause of kidney injury and beyond established biomarkers [3]. In our study, we monitored eGFR decline in KTx recipients over a period of twelve months. However, in contrast to the findings of Zewinger et al., we did not observe a significant association between the uDKK3/crea ratio and eGFR decline in KTx recipients within that time period. 

Meister et al. demonstrated that uDKK3 modulates local T-cell responses [4]. Further, higher uDKK3 levels were associated with a higher histologic degree of IF/TA [3]. Consistent with Meister et al., we found that patients with TCMR or BKPyVAN had the highest uDKK3 levels, but, in parallel, the lowest eGFR values. However, after adjustment for the eGFR, we did not observe an independent association between uDKK3 and the individual histologic entities, and our data failed to establish a causal link between a local T-cell response which is present in T-cellular rejection and BKPyVAN in the transplanted kidney and urinary DKK3 excretion. Furthermore, our findings revealed that there was no independent association between uDKK3 levels and the extent of interstitial fibrosis and tubular atrophy (IF/TA) in KTx recipients. Our analyses indicated that the positive correlation between uDKK3 and high IF/TA degree was also dependent on the eGFR. 

Our findings in the living-donor (LD) cohort support previous observations by Zewinger et al. confirming that there is no correlation between eGFR and uDKK3 in the healthy population [3]. However, we still observed significantly elevated uDKK3 levels one year after Nx, even in the absence of a primary kidney pathology. This raises questions about the intrinsic pathogenic role of uDKK3 [7], as it is substantially increased in patients after Nx, but without causing renal dysfunction or eGFR loss in the following years (as shown in Table 1).

In conclusion, this study points towards a negative correlation between the eGFR and uDKK3. 

Nevertheless, it is important to note that a low uDKK3/crea ratio does not necessarily indicate adequate or good kidney function, as some KTx recipients with CKD stage 4 or 5 have a low uDKK3/crea ratio. The uDKK3/crea ratio is associated with the eGFR to some extent and with outcomes such as allograft failure or specific histologic transplant pathologies such as IF/TA or T-cell-mediated rejection (TCMR) in transplant recipients. However, our findings suggest that the uDKK3/crea ratio is unlikely to be a causal factor.

### Limitations

Due to its observational nature, causal affirmations cannot be made. This study is further limited by the patient recruitment from a single transplant center with a limited number of cases [14]. Moreover, the use of eGFR based on the CKD-EPI formula may have over- or underestimated the patients’ kidney function in some cases. The DKK ELISA kit used was originally validated by the manufacturer for blood samples and cell culture supernatants. However, the results obtained with the kit were reproducible, conclusive in dilution experiments, and are in line with the DKK3 urine values published by others [3,10]. Because the median time since kidney transplantation in the studied patients was 39.8 months, it is not possible for us to make any declarations on the residual function of the native kidneys. Since recipients with residual function of the native diseased kidneys might produce urine with high uDKK3 values and native and transplant kidney urine is mixed, there might have been a measurement bias.

## 4. Materials and Methods

We prospectively collected serum and urine samples from 200 patients who underwent same-day kidney transplant biopsy. In 19 patients, two biopsies were performed. The repeated biopsies of the same patient were excluded from analyses. Indications for biopsy were newly diagnosed donor-specific antibodies in 27 (14.9%) cases, acute or creeping eGFR loss in 105 (58.0%) cases, increasing proteinuria (26, 14.4%), significant BKPyV viremia (>10^4^ cop/mL; 8, 4.4%) or delayed allograft function (need for dialysis within the first week) after KTx (15, 8.3%). Patients with systemic bacterial infections or bacterial urinary tract infections were excluded from the study. To our knowledge, none of the patients suffered from active malignancy.

Data were taken from patients’ files, and personal information was anonymized prior to the analysis. This study was performed in accordance with the Declaration of Helsinki and the International Conference on Harmonization Good Clinical Practice guidelines and approved by the local ethics committee (Ethik Kommission der Ärztekammer Westfalen-Lippe und der Medizinischen Fakultät der Westfälischen Wilhelms-Universität, 2014-381-f-N).

The baseline characteristics of the donors are listed in Table 1, and those of recipients are listed in Table 2.

### 4.1. Living Donors

Eighty-two LDs served as controls, with reduced renal function after Nx but no underlying kidney disease. Urine samples were collected 1 day before Nx and 1–3 days and 1 year after nephrectomy.

We analyzed the urine samples for DKK3 levels or the DKK3/creatinine ratio to investigate whether DKK3 is able to predict the future course of allograft function in the recipient and/or indicate specific circumstances leading to allograft damage. On the other hand, pre-Nx DKK3 was also correlated with donor renal function. The baseline characteristics of the LDs are presented in Table 1.

### 4.2. KTx Recipients

We compared the 144 (79.6%) recipients who had histological acute renal pathology with 37 (20.4%) of the patients without acute histopathology. There were no statistically significant differences in baseline characteristics between the groups. The immunosuppressive regimen consisted of tacrolimus, MMF/MPA, and prednisone in most of the cases. The baseline characteristics of the recipients are summarized in Table 2.

### 4.3. ELISA

Urine concentration of DKK3 was measured by Human DKK-3 DuoSet ELISA Catalog #: DY1118 (R&D Systems, Minneapolis, MN, USA), validated for blood samples and cell culture supernatants, according to the manufacturer’s instructions. The assay range was 31.2–2000 pg/mL with an unspecified sensitivity for DKK-3. Samples above the cut-off value for the assay were re-measured after tenfold dilution in Calibrator Diluent RD6-10 reagent according to the manufacturer’s specifications. The ELISA tests were performed as duplicate determinations.

Urine samples were collected after informed consent within two hours prior to allograft biopsy. From each patient, samples were collected and immediately sent to the research core laboratory. 

Serum was obtained by centrifugation for 10 min at 2000 *g* using a refrigerated centrifuge, transferred into clean polyprolylene tubes, and stored at −80 °C until the time of the assay.

Since the urine concentration of DKK3 depends on the dilution state of the urine, urinary DKK3 levels were normalized to urinary creatinine concentrations [3] and given as the uDKK3/crea ratio.

### 4.4. Statistical Analysis

The data were analyzed using IBM SPSS Statistics 28 (IBM Corp., Armonk, NY, USA). The results are expressed as mean with standard deviation, median with the first and third quartile (IQR), or number/percent. Non-continuous parameters were analyzed using Fisher’s exact test and the chi-square test and continuous parameters were analyzed using the Mann–Whitney U-test and Kruskal-Wallis test, as appropriate. A *p*-value below 0.05 was considered statistically significant.

A cut-off was obtained by ROC analysis and the calculation of the Youden index.

Linear regression analyses and logistic regression analyses, respectively, were used to model the relationship between a dependent variable and one or more explanatory variables.

We analyzed the cumulative incidence of allograft failure after KTx using Kaplan–Meier analysis and the log-rank test. To answer the question of whether uDKK3 is predictive of allograft failure independently of the eGFR, we applied multivariable Cox regression analysis.

### 4.5. Outcome Measures

Kidney biopsies were evaluated by two nephropathologists. Rejection was diagnosed by histological biopsy evaluation based on the current BANFF criteria [15]. Further, IF/TA was graded into three grades, also according to the BANFF classification: Grade I: mild; Grade II: moderate; Grade III: severe. Only patients with representative biopsies were included in the study.

The main clinical outcome measures were kidney function (eGFR calculated using the CKD-EPI equation and urine protein/creatinine ratio (UPCR)) one year after biopsy. In addition, allograft failure was assessed at a mean follow-up time of 39.5 months (±16.8 months). Allograft failure was defined as the definitive re-initiation of dialysis or re-transplantation. Further outcome parameters for the recipients were patient and overall graft survival. Patient survival was defined as the time from RTx to death (from any cause) or last contact for living patients. Overall graft survival was defined as the time from KTx to death (from any cause) or graft failure, whichever occurred first.

Whole blood was analyzed for creatinine (enzymatic assay; Creatinine-Pap, Roche Diagnostics, Mannheim, Germany). Proteinuria was assessed using spot urine for the protein/creatinine ratio (UPCR).

In the LD group, the clinical outcomes were the eGFR and UPCR one, two, and three years after Nx.

## 5. Conclusions

In this study, we provide novel insights into the potential role of uDKK3 as a biomarker for renal function and allograft outcome in KTx recipients. In contrast to the abovementioned studies, which focused on patients with CKD or acute kidney failure, this study shows that uDKK3 levels in kidney transplant patients are not an independent predictor of allograft failure. Rather, they are associated with the eGFR and may reflect the severity of kidney dysfunction to some extent. These findings emphasize the need for further research to explore the role of uDKK3 in kidney function and its potential implications for allograft outcomes.

## Figures and Tables

**Figure 1 ijms-25-05376-f001:**
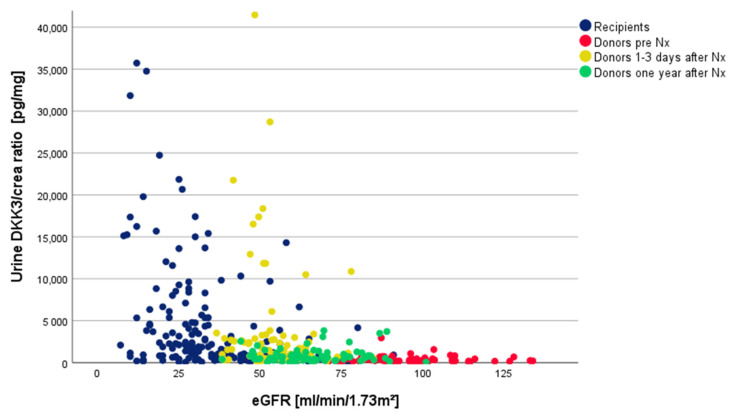
Overview of uDKK3-/crea ratios and the corresponding eGFR values of the donors and recipients at the different measurement times.

**Figure 2 ijms-25-05376-f002:**
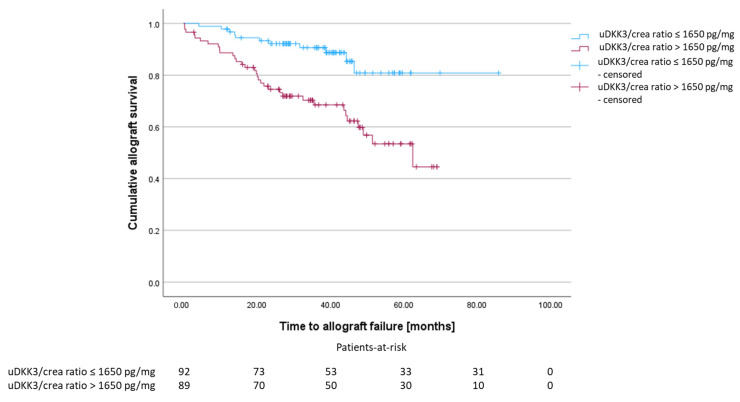
KTx recipients with a uDKK3-/crea ratio > 1650 pg/mg showed a significantly higher incidence of allograft failure during the follow-up from the day of biopsy and uDKK3 measurement (Log rank test *p* < 0.001). Note that the Kaplan–Meier technique does not allow for adjustment. Therefore, see the Cox regression analysis in Appendix A.

**Table 1 ijms-25-05376-t001:** Kidney donors: baseline characteristics and laboratory results.

	All (*n* = 82)
Age at donation, years (min, max)	54.3 (25.9, 69.8)
Sex male n (%)	27 (32.9%)
BMI (kg/m^2^ ± SD)	20.9 (±6.5)
uDKK3/crea ratio pre donation [pg/mg, median (1st, 3rd quartile)]	280 (170, 460)
uDKK3/crea ratio day 1–3 [pg/mg, median (1st, 3rd quartile)]	1490 (580, 2870)
uDKK3/crea ratio day 365 [pg/mg, median (1st, 3rd quartile)]	620 (330, 1170)
eGFR pre donation [mL/min/1.73 m^2^] (±SD)	94.0 (±14.3)
eGFR day 1 [mL/min/1.73 m^2^] (±SD)	54.4 (±10.5)
eGFR day 365 [mL/min/1.73 m^2^] (±SD)	66.5 (±13.5)
eGFR after two years [mL/min/1.73 m^2^] (±SD)	68.5 (±15.6)
eGFR after three years [mL/min/1.73 m^2^] (± SD)	64.3 (±14.0)

BMI: body mass index; SD: standard deviation; uDKK3: urinary dickkopf3; crea: creatinine; eGFR: estimated. Glomerular filtration rate, calculated by CKD-EPI formula.

**Table 2 ijms-25-05376-t002:** Baseline characteristics and outcomes of the recipients.

	All (*n* = 181)	Acute Histologic Pathology (*n* = 144, 79.6%)	No Acute Histologic Pathology (*n* = 37, 20.4%)	*p*-Value
Age (years, mean ± SD)	51.2 (±15.6)	51.0 (± 16.2)	52.0 (± 12.1)	0.782
Sex male, n (%)	121 (66.9%)	100	21	0.117
Time since KTx (months, median (1st, 3rd quartile)	39.8 (8.7, 108.0)	46.1 (7.5, 110.6)	33.0 (10.9, 99.9)	0.177
Immunosupressive regimen n, (%)				
Tacrolimus	133 (73.5%)	105 (72.9%)	28 (75.7%)	0.836
Cyclosporine	29 (16.0%)	22 (15.3%)	7 (18.9%)	0.618
Everolimus	40 (22.1%)	36 (25.0%)	4 (10.8%)	0.076
MMF/MPA	143 (79.0%)	111 (77.1%)	32 (86.5%)	0.262
Steroids	174 (96.1%)	138 (95.8%)	36 (97.3%)	1.000
Belatacept	4 (2.3%)	4 (2.8%)	0 (0%)	0.583
uDKK3/crea ratio (pg/mL, median (1st, 3rd quartile)	1552 (661, 5335)	2050 (710, 6610)	910 (510, 2360)	0.008
eGFR one year after biopsy (mL/min/1.73 m^2^, mean ± SD)	38.3 (±18.1)	37.1 (± 18.3)	42.9 (± 16.9)	0.085
UPCR at biopsy (median (1st, 3rd quartile))	203 (89, 581)	239 (103, 910)	112 (64, 210)	<0.001
UPCR one year after biopsy (median (1st, 3rd quartile))	195 (85, 494)	225 (92, 528)	120 (62, 263)	0.007
CMV DNAemia at biopsy, n (%)	10 (5.5%)	6 (4.2%)	4 (10.8%)	0.123
BKPyV DNAemia at biopsy, n (%)	20 (11.0%)	19 (13.2%)	1 (2.7%)	0.081
IF/TA *				0.007
mild	122 (67.4%)	90 (62.5%)	32 (86.5%)	
moderate	35 (19.3%)	31 (21.5%)	4 (10.8%)	
severe	20 (11.0%)	20 (13.9%)	0 (0%)	
missing values	4 (2.2%)	3 (2.0%)	1 (2.7%)	

KTx: kidney transplantation, MMF/MPA: mycophenolate-mofetil/mycophenolic acid; uDKK3: urinary dickkopf3; crea: creatinine; eGFR: estimated glomerular filtration rate, calculated by CKD-EPI formula; UPCR: urine protein creatinine ratio; CMV: cytomegalovirus; BKPyV: BK-polyomavirus; IF/TA: interstitial fibrosis and tubular atrophy. * missing n: no IF/TA classification available.

**Table 3 ijms-25-05376-t003:** Multivariable linear regression analysis for the dependent variable eGFR at one year.

Variable	Regression-Coefficient	95% CI	Beta	*p* Value
uDKK3/crea ratio on biopsy day	0.000	−0.012–0.013	0.004	0.941
eGFR on biopsy day	0.815	0.694–0.936	0.738	<0.001

uDKK3: urinary dickkopf3; crea: creatinine; eGFR: estimated glomerular filtration rate, calculated by CKD-EPI formula.

**Table 4 ijms-25-05376-t004:** The histological diagnoses underlying the biopsies.

Diagnosis	*n*, %
Antibody-mediated rejection (AMR)	18, 19.9%
T-cellular rejection (TCMR)	12, 6.6%
Combined rejection	43, 23.8%
T-cell-borderline rejection	16, 8.8%
BKPyV nephropathy (BKPyVAN)	9, 5.0%
Ischemia reperfusion injury (IRI)	17, 9.4%
Severe chronic injury	13, 7.1%
Calcineurin-inhibitor toxicity	7, 3.9%
Recurrent glomerular disease	7, 3.9%
Bland with no acute histological pathology	37, 20.4%
adenoviral nephritis	2, 1.1%

BKPyV: BK polyomavirus.

## Data Availability

The datasets generated during and/or analyzed during the current study are available from the corresponding author on reasonable request.

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
