# Peer review of "Urinary Dickkopf 3 Is Not an Independent Risk Factor in a Cohort of Kidney Transplant Recipients and Living Donors"

_ijms, 2024, doi:10.3390/ijms25105376_

Round 1

Reviewer 1 Report

Comments and Suggestions for Authors

First the evaluation of this marker in kidney transplant patients is a valuable idea.

This paper has several inconsistences in the analysis of this group of patients.

The authors use a cohort of patients with kidney biopsy. We have no information about the quality of biopsy. Were they all representative? What was the degree of tubulitis?

The histological diagnosis are very different, so the involvement of tubular compartment must also be.

Was there need of aditional immunosuppressive treatment or other?

Were all patients a first kidney transplant? What were the characteristics of the donor? Type of dead, age, expanded criteria?

Any complications immediately after kidney transplant? Rejection? Infection?

There is no reference to the cause of kidney failure. What about the comorbilities?

The abcense of several informations in a group of patients who appear very different does not allow for any conclusion.

Author Response

Please find our response as attached reply letter.

Kindest regards.

Reviewer 2 Report

Comments and Suggestions for Authors

Dear authors than you very much for giving me the opportunity to read your work. You describe the association between uDKK3 and renal function as depicted by eGFR in two separate patient subsets: Living kidney donors and Kidney recipients. I have the following comments: 

1. What was the residual kidney function in renal transplant recipients? Since native kidneys are damaged, recipients with residual kidney function produce urine with high uDKK3 values. Native and transplant urine are collected in the same urinary bladder, therefore there is a measurement bias. Ideally urinary DKK3 should be measured before NTx in recipients with residual renal function.  Even then, there is no absolute method to adequately identify uDKK3 produced from native or transplant kidney the months following renal transplantation. Which were uDKK3 values in anuric kidney recipients within your cohort after NTx? Is there any predictive function of higher uDKK3 and renal function evolution? Is it possible to stratify data according to residual renal function in recipients with residual renal function? Is there any difference? 

2. Living kidney donors are selected in the abscence of major health issues. Yet studies concerning living kidney donation (for example Muzalee et al JAMA 2014;311(6):579-86) and eGFR deterioration often require an observation time of more than 5 years. In your cohort observation time was 1 year. Smaller observation time, might underpower the capacity of  utilizing uDKK3 as a predictor of renal function evolution. Therefore the phrase in lines 244-246 is not accurate.

3. After normalizing for the above confounders I would depict uDKK3 and eGFR association in a different manner. In the same graph individual values of uDKK3 and eGFR at the beginning and at the end of the observation period. Then define for example quartiles/tertiles of individual uDKK3 values. Do uDKK3 values in the higher quartile/tertile at the beginning individually predict lower eGFR values? 

All the best. 

Author Response

Please find our responses in the attached reply letter.

Kindest regards! 

Reviewer 3 Report

Comments and Suggestions for Authors

congratulations for your smart study,The design is very original  and the results are sound.

In my opinion the only problem about your manuscrit is the reiteration of showing the results.You insist on the relathionship between uDKK3· and GRF repetitively in the discussion ,As a matter of fact you should omit several refecences  in the text and only show the relathionship once

thank you

Author Response

Please find our responses in the attached cover letter.

Kindest regards.

Reviewer 4 Report

Comments and Suggestions for Authors

The article titled: Urinary dickkopf 3 is not an independent risk factor in a cohort 2 of kidney transplant recipients and living donors, the subject of the article is based on the assessment of the role of Urinary Dickkopf 3 (uDKK3) as a potential biomarker in the assessment of the progression of chronic kidney disease.

The article has a typical structure of a scientific publication, and the topic itself is interesting; certainly, conducting research with patients increases the article's value. However, before the work is published, I would like to clarify a few issues that emerged during the analysis of the article, which I present below:

- The basis of the study are patients and biopsies taken from them, can the authors describe in more detail the inclusion criteria and exclusion of study patients and the control group. Whether these people had any other additional factors, such as cancer, autoimmune diseases, or infections resulting from the immunosuppressive drugs taken.

- Did the authors consider how the immunosuppression may affect the biomarker being studied? Are such studies published, or can it be clearly said that the drugs taken do not affect the functions of kidney tubular epithelial cells?

- When it comes to ELISA tests, how many repetitions were performed?

- In the text itself, we have incorrect editing of citations which should look like this [1]

- Table 4 is missing an entry for severe chronic injury.

- The numbering of subchapters should also be corrected, with appropriate numbers added.

Additional materials—When I analyze additional materials, figures 1 and 2 show statistical significance in the form * and °. I ask the authors to provide signatures explaining what these symbols mean; this will make it easier for the reader to read the work.

To sum up, the study is exciting and develops existing theses related to (uDKK3). What is certainly important to emphasize is that at the moment, this marker can only be an additional molecule, which, together with other parameters, can only help in some way in adapting the therapy and treating the patient.

Nevertheless, the authors did a good job, in my opinion.

Author Response

Please find our responses in the attached cover letter.

Kindest regards!

Reviewer 5 Report

Comments and Suggestions for Authors

The manuscript aimed to investigated Urinary Dickkopf 3 (uDKK3) as a marker for future function of transplanted kidney. The authors studied uDKK3 kinetics before and after unilateral nephrectomy in leaving donors as well as in a cohort of kidney transplant recipients undergoing renal allograft biopsy. The study is interesting and brings novelty, however, I suggest rewriting the manuscript as in the current form it is difficult to understand. There are numerous results which in some cases exclude each other (see comments). My comments are listed below:

·         I suggest adding ‘range’ in the brackets with numbers (e.g. lines 81 and 82: (25.9, 69.8) and (170, 460) etc.) as I suppose that the numbers represent ranges for mean values given beforehand. Moreover, I suggest giving standard deviation when mean value is given and range for median.

·         The abbreviation ‘uDKK3/crea’ was not explained. Did the authors mean ‘uDKK3/creatinine’ as in the Abstract?

·         Please, explain more clearly whether there was or not the correlation between uDKK3/crea ad eGFR as the sentence: ‘Although 146 uDKK3/crea ratio at biopsy was significantly correlated to eGFR values one year after biopsy in univariable analysis (p=0.002, r= 0.000, beta 0.232), there was no independent association between uDKK3/crea-ratio and eGFR one year after biopsy (p=0.941, r=0.000, 149 beta 0.004).’ is confusing and in my opinion the correlation was and was not observed at the same time. Both analysis concern the results one year after biopsy.

·         Did the authors analyze the influence of immunosuppressive drug on the results of the study? Are there any data concerning the association between uDKK3/crea ratio and the immunosuppressive drug?

·         There are too many tables (n=6) in this manuscript. For example, Table 4 does not need to be in the main text. The data may be added to one of the existing table or moved to Supplement. Moreover, the figures and tables are cited in a places distant from its position in the text what causes confusion during reading.

·         Why is the references list doubled?

Comments on the Quality of English Language

·         Minor language correction is needed.

Author Response

(The authors gave the same response as above.)

Round 2

Reviewer 1 Report

Comments and Suggestions for Authors

Once again, this study, in wich concern kidney transplant patients, was not well designed.

The authors refer to another study with KT patients, Dickkopf 3—A New Indicator for the Deterioration of Allograft Function After Kidney Transplantation, Front Med (Lausanne). 2022; 9: 885018.

The Front Med authors refer to the importance of immunosuppression.

“ Regarding the impact of immunosuppressives on the development of DKK3, no further data are available. But in literature, the influence of DKK3 on T- lymphocytes is discussed. As already mentioned, DKK3 seems to trigger a profibrotic T cell response. Federico et al. were also able to show that after an antibody-mediated blockade of DKK3, an increased presence of protective T cells (IFNγ-producing Th1 and Tregs) can be demonstrated (13). Taking this into account, the evidence of lower DKK3 values under a T-cell depleting therapy seems understandable. Further investigations on the influence of immunosuppressives would be useful to further evaluate specific “anti-DKK3 and therefore presumable anti-fibrotic immunosuppressive protocols.”

The methodology was done taking into account this “ Each recipient was grouped according to its underlying immunological risk profile (CDC-PRA, DSA, etc.) before transplantation and thereafter treated by a pre-defined immunological algorithm “.

In opposition, in this study, the authors bring together a very different group of patients, not taking into account many variables of great importance in KT, already refered in first revision.

In Front Med study DKK3 was measured non-invasively in the urine 14 days, 3, 12, 18, 24, 30, and 36 months postTx. In this study, the authors have only one determination, in patients with different KT times.

21 patients with BKPyVAN and TCMR, what is the statistical value?

It is not possible have an interpretation in this group of patients.

Reviewer 2 Report

Comments and Suggestions for Authors

Dear authors, thank you very much for this improved version of your work. You addressed the majority of the concerns raised. I still have the following questions/suggestions concerning the following statement: 

“Which were uDKK3 values in anuric kidney recipients within your cohort after NTx?” -> Since we measured uDKK3 in urine samples, we have no information on uDKK3 in anuric patients.

1. Can you detect anuric patients before kidney transplantation within your cohort? If you do not have this capacity, please state this in the Limitations section of your manuscript.

2. If you can detect anuric patients before renal transplantation: These patients acquire the capacity for urine production after renal transplantation. Therefore measured uDKK3 originates only from the transplanted kidney. How is uDKK3 correlated with eGFR evolution in this patient subgroup? Could you detect a prognostic significance?

3. The title should be concise: Urinary dickkopf 3 is not an independent risk factor in a cohort of kidney transplant recipients and living donors. The title should indicate the context in which this biomarker is a risk factor (e.g. dickkopf 3 is not an independent risk factor e.g. for... CVD, acute kidney injury, chronic kidney disease progression in a cohort...).

All the best.

Reviewer 4 Report

Comments and Suggestions for Authors

Dear Authors,

 Thank you for considering the suggestion, the article looks much better in my opinion now. The subject of the research is very interesting, and examining the aspect of the occurrence of uDKK3 in the context of patient prognosis is necessary. Thank you also for your answer regarding the potential impact of medications, I hope that this area will also be of interest to you.

 To sum up, the authors' answers are satisfactory, and the study itself is interesting and may be published.